# Comprehensive Knowledge Distillation with Causal Intervention

**Xiang Deng**
Computer Science Department
State University of New York at Binghamton
xdeng7@binghamton.edu

**Zhongfei Zhang**
Computer Science Department
State University of New York at Binghamton
zhongfei@cs.binghamton.edu

## Abstract

Knowledge distillation (KD) addresses model compression by distilling knowledge from a large model (teacher) to a smaller one (student). The existing distillation approaches mainly focus on using different criteria to align the sample representations learned by the student and the teacher, while they fail to transfer the class representations. Good class representations can benefit the sample representation learning by shaping the sample representation distribution. On the other hand, the existing approaches enforce the student to fully imitate the teacher while ignoring the fact that the teacher is typically not perfect. Although the teacher has learned rich and powerful representations, it also contains unignorable bias knowledge which is usually induced by the context prior (e.g., background) in the training data. To address these two issues, in this paper, we propose comprehensive, interventional distillation (CID) that captures both sample and class representations from the teacher while removing the bias with causal intervention. Different from the existing literature that uses the softened logits of the teacher as the training targets, CID considers the softened logits as the context information of an image, which is further used to remove the biased knowledge based on causal inference. Keeping the good representations while removing the bad bias enables CID to have a better generalization ability on test data and a better transferability across different datasets against the existing state-of-the-art approaches, which is demonstrated by extensive experiments on several benchmark datasets[1].

## 1  Introduction

The superior performances of deep neural networks (DNNs) are accompanied with large amounts of memory and computation requirements, which seriously restricts their deployment on resource-limited devices. An effective and widely used solution to this issue is knowledge distillation [19, 37] that compresses a large network (teacher) to a compact and fast network (student) by knowledge transfer. To this end, the student obtains a significant performance boost.

The original knowledge distillation (KD) [19] uses the softened logits generated by a teacher as the targets to train a student. Ever since then, substantial efforts including [37, 45] have been made on aligning the sample representations learned by the student with those learned by the teacher using different criteria. However, almost all the existing approaches [45, 49, 58] have overlooked the class representations. Good class representations are beneficial to sample representation learning, since they can shape the sample representation distribution. To address this issue, we propose comprehensive distillation to incorporate the class representations learned by the teacher into the distillation process.

On the other hand, as the teacher has learned rich and powerful representations, the existing approaches enforce the student to fully mimic the behavior of the teacher. However, fully imitating

---

[1]Code: https://github.com/Xiang-Deng-DL/CID

35th Conference on Neural Information Processing Systems (NeurIPS 2021).

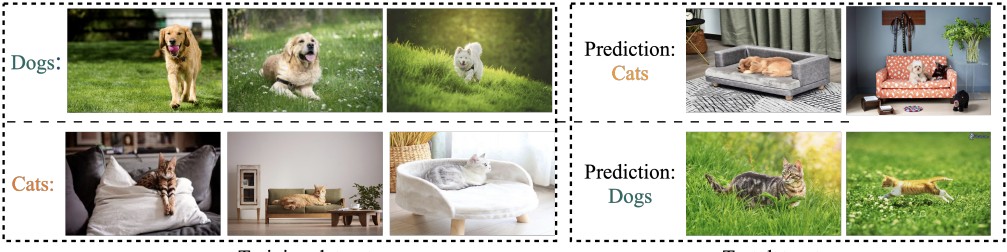

Figure 1: Misclassification caused by context prior in the training dataset.

the representations of the teacher may not be optimal, since the bias contained in the teacher is also transferred to the student. The bias is usually caused by the context prior in the training data. For example, as shown in Figure 1, the dogs in the training dataset are usually on green grasses and the cats are in a room, which misleads the trained classifier to classify the cats on green grasses in the test dataset as dogs and the dogs in a room as cats due to the bias induced by the context. Similar cases can also happen on the attributes of data samples, e.g., when the colors of the dogs in the training dataset are mostly black, the black cats in the test dataset may be wrongly classified as dogs. Transferring this kind of the bias contained in the pretrained teacher to the student hurts the student.

Since the biased knowledge in the teacher is caused by the training data, we assume that the training data used by the teacher and those used by the student are from the same distribution. This is not a strong assumption in knowledge distillation literature as almost all the existing work uses the same dataset when training a teacher and a student, which obviously satisfies the assumption. Contrary to this assumption, when the training data for the teacher and the student are from different distributions, two issues arise. First, the teacher may not be able to teach the student anymore due to the data distribution discrepancy. Second, new biases will be introduced in the distillation process from the new training dataset. We leave these questions for the future work.

Under the above assumption, we formulate the causal relationships [32] among the pretrained teacher, the samples, and the prediction in a causal graph as shown in Figure 4(a). More details are given in Section 3.2.1. We then use the softened logits learned by the teacher as the context information of an image to remove the biased knowledge based on backdoor adjustment [14]. To this end, we propose a simple yet effective framework (i.e., CID) to achieve comprehensive distillation and bias removal.

We summarize our contributions and the differences from the existing approaches as follows:

- We propose a novel knowledge distillation framework, i.e., CID, which captures comprehensive representations from the teacher while removing the bias with causal intervention. To our best knowledge, this is the first work to study how to use causal inference to address KD-based model compression.

- CID is different from the existing approaches in two aspects. First, CID is able to transfer the class representations which are largely ignored by the existing literature. Second, CID uses softened logits as sample context information to remove biases with causal intervention, which differs from the existing literature that uses the softened logits as the training targets to train a student. Keeping the good knowledge while removing the bad bias enables CID to have a better generalization on test data and a better transferability on new datasets.

- Extensive experiments on several benchmark datasets demonstrate that CID outperforms the state-of-the-art approaches significantly in terms of generalization and transferability.

## 2   Related Work

**Knowledge Distillation.** Hinton et al. [19] propose the original KD that trains a student by using the softened logits of a teacher as targets. Compared to one-hot labels, the logits provide extra information learned by the teacher [19, 13]. However, KD fails to transfer the powerful representations learned by the teacher. Ever since then, many efforts have been made on aligning the sample representations learned by a student and a teacher. FitNet [37] aligns the sample representations learned by a student with those learned by a teacher through regressions. AT [56] distills sample feature attention from a teacher into a student. CRD [45] maximizes the mutual information between sample representations

learned by a student and a teacher. SRRL [21] aligns the sample representations of a teacher and a student by using the teacher's classifier. CC [35] and SP [46] transfer the sample correlation over the whole dataset to the student, which may contain redundant and irrelevant information as pointed out in [7]. Similarly, other approaches [52, 20, 22, 25, 42, 48, 17, 8, 2, 23, 1, 40, 45, 49, 21, 12] use different criteria to align the sample representations. We notice that almost all these approaches only transfer the sample representations while largely ignoring the class representations which can benefit the sample representation learning by shaping the sample representation distribution.

**Causal Inference.** Causal inference [33, 34, 38] aims to explore the cause-effect relationships between different variables. It can not only be used to interpret a particular phenomenon [6, 26], but also serve as a tool to address problems by determining and using the causal effects [3, 5, 29]. Recently, it has been introduced to machine learning [4] and has been used in different applications, including but not limited to domain adaptation [15, 27], imitation learning [10], image captioning [51], scene graph generation [44], visaul dialog [36], few-shot learning [54], imbalance classification [43], semantic segmentation [53, 57], VQA [28], and unsupervised learning [47]. In this work, we provide an interventional framework for knowledge distillation to remove the biased knowledge in the teacher.

## 3 Comprehensive, Interventional Distillation

In this section, we first describe the comprehensive distillation which takes both the sample and class representations into account. We then present interventional distillation to remove the biased knowledge with causal intervention and thus achieve comprehensive, interventional distillation (CID).

### 3.1 Comprehensive Representation Distillation

CID considers both sample and class representations and thus achieves comprehensive distillation.

**Which layer's sample representations are transferred?** Many approaches [37, 45] transfer the intermediate or the last few layers' feature representations. In contrast, CID only distills the feature vectors in the last layer (before the classifier), since only these vectors are directly involved in making the final prediction. We empirically validate this point in Section 4.1.

**The criterion to align representations.** We slightly modify the mean square error (MSE) as the criterion to align the sample representations. MSE is used in FitNet [37] for transferring the sample representations. However, we find that MSE has a disadvantage that it biases towards the samples that have large-norm features. For example, for $\text{MSE}(V_a, V_b)=||V_a - V_b||^2$ where $V_a$ and $V_b$ are the variable and target, respectively, when $V_a = [0.02, 0.02]$ and $V_b = [0.01, -0.01]$, the gradient is only $2(V_a - V_b) = [0.02, 0.06]$. It is observed that $V_a$ and $V_b$ have totally different directions and are orthogonal, but the gradient is very small due to the small norms of the two vectors. In contrast, for large vectors $V_a = [10, 20]$ and $V_b = [9.5, 19.5]$, the gradient is [1, 1] which is much larger than [0.02, 0.06], even if the two vectors are very close in terms of both directions and norms. The inherent disadvantage of MSE makes the samples with different feature norms contribute differently to the student, which induces biases. As shown in Figure 2,

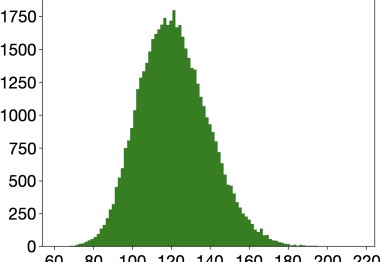

Figure 2: Feature norm distribution of ResNet32×4 on CIFAR-100.

the norms of the sample features learned by the teacher are across a wide range and have a noticeable variance. One natural idea to address this issue is to use the MSE of the normalized vectors, i.e., $||\frac{V_a}{||V_a||_2} - \frac{V_b}{||V_b||_2}||^2$. However, this loss only aligns the directions of the two vectors and the minimum point is not necessarily $V_a = V_b$ anymore, e.g., $V_a = [1, 1]$ and $V_b = [100, 100]$ are a solution to this loss due to the same vector direction although they are different substantially. To address this issue, we propose Normalized MSE (NM_MSE):

$$\mathcal{M}(V_a, V_b) = \frac{\text{MSE}(V_a, V_b)}{||V_b||^2} = \frac{||V_a - V_b||^2}{||V_b||^2} \tag{1}$$

where $||.||^2$ denotes the square of $L_2$ norm. NM_MSE can be considered as a sample-wise weighted MSE and the weights $\frac{1}{||V_b||^2}$ for different samples are negatively related to their target feature vector

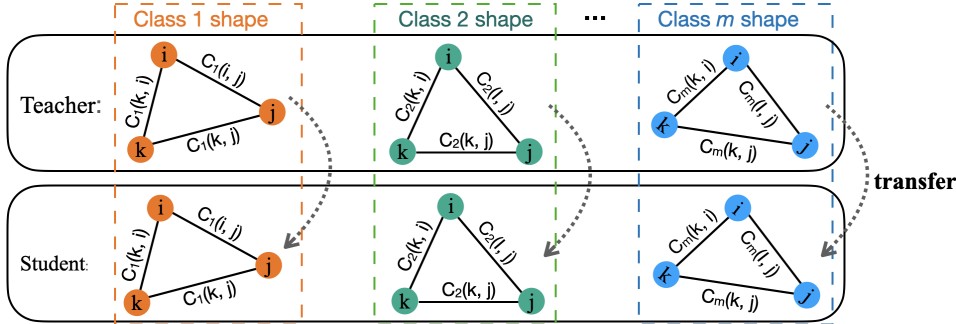

Figure 3: Class representation transfer.

norms, which mitigates the bias and makes different samples equally contribute to the student. Although this is a slight modification to the original MSE, we empirically find that it performs much better than MSE on benchmark datasets.

**Class representations.** The existing approaches only enable the student to capture the sample representations from the teacher while overlooking the class representations. CID incorporates this part into the distillation process. CID uses the class shapes to represent the class representations. As shown in Figure 3, the shape of a class is a graph with all the samples in the class as the nodes. The weight between two nodes is defined as the representation similarity between the two nodes. We adopt the cosine similarity and thus the class representation of class $i$ is expressed as:

$$C_i = [\frac{h_1^i}{||h_1^i||_2}, \frac{h_2^i}{||h_2^i||_2}, ..., \frac{h_k^i}{||h_k^i||_2}]^T [\frac{h_1^i}{||h_1^i||_2}, \frac{h_2^i}{||h_2^i||_2}, ..., \frac{h_k^i}{||h_k^i||_2}] \qquad (2)$$

where $h_j^i$ denotes the feature vector of sample $j$ in class $i$; $k$ is the total number of samples in class $i$; superscript $^T$ means transpose; $||.||_2$ denotes $L_2$ norm. The class representation is different from the sample-class relation defined in [7] which only captures sample-to-class-center similarity while failing to capture sample-to-sample relations and thus cannot well represent a class.

**The objective for comprehensive representation distillation.** CID transfers comprehensive knowledge consisting of both sample and class representations from a teacher to a student, and thus the objective for comprehensive distillation is written as:

$$\mathcal{L}_{rep}(S, T) = \alpha \mathcal{M}(h_s^T W, h_t) + \beta \sum_{i=1}^{m} ||C_i^S - C_i^T||^2 \qquad (3)$$

where $\mathcal{M}$ is the proposed NM_MSE; $h_s \in \mathbb{R}^{m_s}$ and $h_t \in \mathbb{R}^{m_t}$ are the sample representations learned by student S and teacher T, respectively; $W \in \mathbb{R}^{m_s \times m_t}$ is a linear transformation for converting $h_s$ to the space with dimension $m_t$; $C_i^S$ and $C_i^T$ are the $i$th class representations learned by S and T, respectively; $m$ is the total number of classes; $\alpha$ and $\beta$ are two balancing weights.

## 3.2 Interventional Distillation

Although a teacher has learned good representations, it is typically not perfect. Comprehensive distillation enables the student to inherit the superior representations from the teacher while it also introduces the bias to the student. To address this issue, we use causal intervention to remove the bias.

### 3.2.1 Structural Causal Model

In knowledge distillation, the pretrained teacher with the context information in training data can be considered as the prior knowledge for training the student. We illustrate the causalities among prior knowledge $K$, sample $X$, and prediction $Y$ in Figure 4(a), where $A \rightarrow B$ denotes that $A$ is the causer of $B$. We describe the causal relationships among these variables in the following.

$K \rightarrow X$: the context prior in $K$ determines where the object appears in an image $X$, e.g., the context prior in the training dataset in Figure 1 puts the dog object in green grasses instead of rooms.

$K \rightarrow J \leftarrow X$: $J$ is the context-based representation of $X$ by using the context bases in $K$. This relationship exists due to the fact that even for the same image, its context representation under different dataset contexts or with different pretrained teachers can differ substantially.

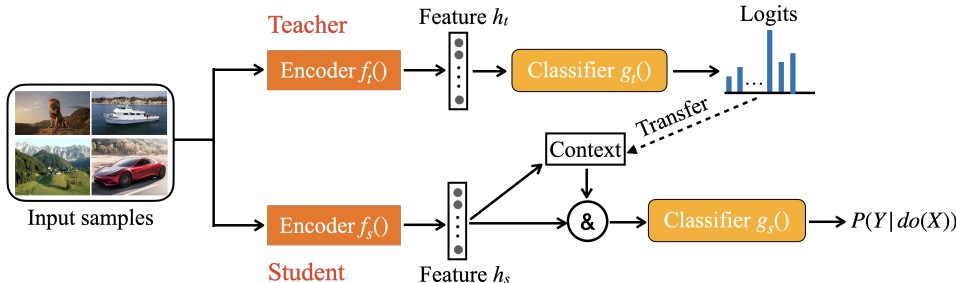

Figure 5: Interventional Distillation. A network can be represented as an encoder $f()$ followed by a linear classifier $g()$ so that teacher $T(X) = g_t(f_t(X))$ and student $S(X) = g_s(f_s(X))$.

$X \rightarrow Y \leftarrow J$: Besides the regular $X \rightarrow Y$, the prediction is also affected by the prior knowledge $K$ through mediation $J$. For example, in Figure 1, the cats in the test dataset are misclassified to dogs, since the context prior in $K$ misleads the model to focus on the grass feature in $X$.

Therefore, the prior knowledge $K$ is a confounder of $X$ and $Y$. The existing approaches that directly learn $P(Y|X)$ from the teacher bring the bias to the student model. We propose to model $P((Y|do(X))$ with causal intervention [33] to remove the bias.

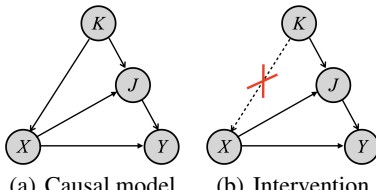

(a) Causal model    (b) Intervention

Figure 4: Causal model and intervention.

### 3.2.2 Interventional Distillation via Backdoor Adjustment

After determining the cause-effect relationships, we use causal intervention $P(Y|do(X))$ instead of $P(Y|X)$ as the classifier, which pursues the true causality from $X$ to $Y$ by removing the effects of confounder $K$. Physical intervention, i.e, collecting samples with objects in all possible contexts evenly, is impossible [57]. Thanks to backdoor adjustment, we can model $P(Y|do(X))$ by cutting off $K \rightarrow X$, which is achieved by stratifying the confounder into pieces $K = \{k_1, k_2, ..., k_{|K|}\}$, so that $K$ is not a confounder of $X$ and $Y$ anymore as shown in Figure 4(b). The de-confounded student is expressed as:

$$P(Y|do(X)) = \sum_{i=1}^{|K|} [P(Y|X, J = g(X, k_i))P(k_i)] \tag{4}$$

where g() is a function which we define later for generating context representation $J$ from X and $k_i$.

As there are $m$ classes which can be considered as $m$ different context items [54], we set each item $k_i$ of the prior knowledge to a class $\mathbf{c}_i$, i.e., $K = \{\mathbf{c}_i\}_{i=1}^m$. The $m$ context base vectors are set to the class centers. Since different samples in a class have different probabilities of containing the object $\mathbf{c}_i$, we use the the weighted average of sample features as the class center.

$$\bar{c}_i = \frac{\sum_{j=1}^k P(\mathbf{c}_i|x_j)h_j}{\sum_{j=1}^k P(\mathbf{c}_i|x_j)} \tag{5}$$

where $k$ is the total number of samples in class $\mathbf{c}_i$. $P(\mathbf{c}_i|x_j)$ is set to the teacher learned probability.

With context base vectors, we define the sample-specific context representation $J$ as a linear combination of the context base vectors. As the logits learned by the teacher contain sample-to-class similarities, we use the softened logits to approximate the context coefficients to provide context information. The coefficient learned by the teacher for sample $X$ on base $\bar{c}_i$ is written as $a_i^t = \sigma(\frac{T(X)}{\tau})[i]$, where $\tau$ is temperature to soften the logits and $\sigma$ is the softmax function. $J$ can thus be expressed as: $J = g(X, \mathbf{c}_i) = a_i \bar{c}_i$. Since the teacher has learned appropriate context information, we enforce the student to learn the context information from the teacher, which leads to the final interventional distillation objective:

$$\mathcal{L}_{inv} = P(Y|do(X)) + \mathcal{K}(a_i^s, a_i^t) = \sum_{i=1}^m [P(Y|f_s(X)\&(a_i^s \bar{c}_i))P(\mathbf{c}_i)] + \mathcal{K}(a_i^s, a_i^t) \tag{6}$$

where $\&$ denotes the concatenation operation; $\mathcal{K}$ is a metric to force the student to learn the context information from teacher. The learned context information is then used in the first term (for causal intervention) to make the final prediction through a linear classifier. we set $\mathcal{K}$ to KL-divergence in this paper. We simply set the weight for $\mathcal{K}()$ to 1 as we find that it works very well. $P(\mathbf{c}_i)$ is set to the percentage of the samples in class $\mathbf{c}_i$, e.g., in balanced datasets, $P(\mathbf{c}_i) = \frac{1}{m}$. We summarize the interventional distillation in Figure 5.

The $\sum$ operation in (6) makes the forward cost of the final linear classifier linearly increase with the number of classes. This issue can be addressed by adopting the normalized weighted geometric mean [50] as an approximation:

$$\mathcal{L}_{inv} \approx P(Y|f_s(X) \& \sum_{i=1}^{m}[P(\mathbf{c}_i)a_i^s \bar{c}_i]) + \mathcal{K}(a_i^s, a_i^t) \tag{7}$$

**The complete objective of CID.** CID aims to achieve comprehensive knowledge distillation while removing the bad bias with causal intervention. Thus, its final objective is written as:

$$\mathcal{L}_{CID} = \mathcal{L}_{inv} + \alpha \mathcal{M}(h_s^T W, h_t) + \beta \sum_{i=1}^{m} ||C_i^S - C_i^T||_2 \tag{8}$$

## 4 Experiments

In this section, we first conduct ablation studies and then compare CID with SOTA approaches.

### 4.1 Ablation Studies

The ablation studies are conducted on CIFAR-100 by using WRN-40-2 and WRN-16-2 as the teacher and the student, respectively.

**Effects of different components of CID.** We use $w/o$ SR, $w/o$ CR, and $w/o$ INV to denote CID without the sample representation distillation, without the class representation distillation, and without the intervention $P(Y|do(X))$ by using the regular $P(Y|X)$, respectively. As shown in Figure 6, the performances drop significantly without any of these terms. Specifically, as expected, sample representations play the most important role with performance gain 1.51%, since sample representations are directly involved in making final predictions and are further used in class representation distillation and causal intervention for removing biases. The improvement of the class representation distillation is 0.65%, which suggests that good class presentations benefit the sample representation learning and thus benefit the performance. On the other hand, by using the proposed $P(Y|do(X))$ instead of $P(Y|X)$ to pursue the true causality from $X$ to $Y$, the student obtains 0.70% improvement, which demonstrates the effectiveness and necessity of the interventional distillation.

**Effects of NM_MSE.** The proposed NM_MSE is modified from MSE by using a personalized weight for each sample to remove the feature norm biases of the teacher so that each sample contributes equally to the student model. Despite its simplicity, it is observed in Figure 6 that NM_MSE (i.e., CID) substantially outperforms MSE (i.e., CID$_{mse}$), which demonstrates the superiority of NM_MSE.

**Which layer's representations should be transferred?** CID transfers the last layer's feature vectors of the teacher to the student with the motivation that these features are directly involved in making the final prediction. We check the effects of distilling the representations in different layers. We report the results in Figure 7. It is observed that the sample representations in the last layer are more effective than those in the other layers, and even better than the combination of the representations in all the layers, The reason can be that as the representations in the intermediate layers are not directly used for the final prediction, enforcing the student to imitate these representations hurts the learning ability and flexibility of the student which has a small capacity.

### 4.2 Comparison Settings with SOTA Approaches

We compare CID with SOTA approaches across varieties of **(a)** benchmark datasets (i.e., CIFAR-10 [24], CIFAR-100 [24], Tiny ImageNet [2], and ImageNet [11]), **(b)** network architectures (i.e., ResNet

---
[2]https://tiny-imagenet.herokuapp.com

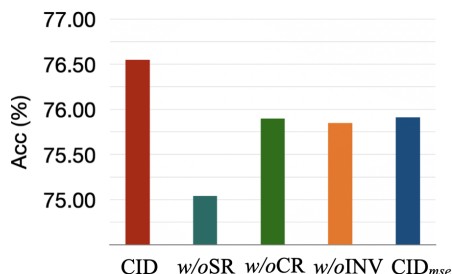

Figure 6: Effects of different components of CID.

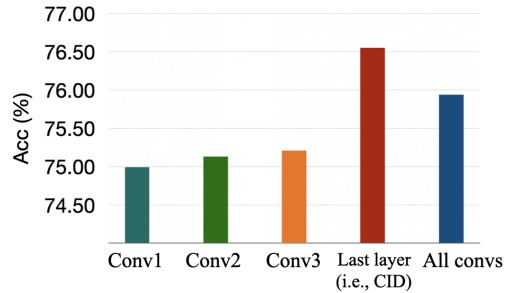

Figure 7: Effects of distilling the sample representations in different layers.

Table 1: Test accuracies (%) on CIFAR-10.

| Teacher (#Params) | WRN-16-4 (2.75M) | ResNet-56 (0.86M) | ResNet-56 (0.86M) | WRN-16-4 (2.75M) | ResNet-110 (1.73M) |
|---|---|---|---|---|---|
| Student (#Params) | WRN-16-1 (0.18M) | ResNet-14 (0.18M) | ResNet-8 (0.08M) | ResNet-14 (0.18M) | WRN-16-1 (0.18M) |
| Teacher | 95.04 | 93.87 | 93.87 | 95.04 | 94.00 |
| Vanilla Student | 91.32 | 91.33 | 88.55 | 91.33 | 91.32 |
| KD | 92.55±0.10 | 91.88±0.22 | 88.70±0.18 | 92.33±0.23 | 91.52±0.16 |
| FitNet | 92.51±0.26 | 91.74±0.19 | 88.74±0.15 | 92.55±0.14 | 91.46±0.17 |
| CC | 92.54±0.23 | 92.09±0.27 | 88.71±0.16 | 92.54±0.18 | 91.66±0.29 |
| RKD | 92.77±0.18 | 92.10±0.14 | 88.14±0.08 | 92.60±0.29 | 91.81±0.21 |
| AB | 92.39±0.30 | 92.14±0.19 | 88.85±0.18 | 92.40±0.09 | 91.31±0.21 |
| CRD | 90.96±0.20 | 90.41±0.13 | 88.40±0.09 | 91.17±0.14 | 90.27±0.20 |
| SRRL | 92.56±0.14 | 91.87±0.12 | 88.76±0.20 | 92.30±0.25 | 91.83±0.15 |
| CID | **92.95±0.10** | **92.31±0.20** | **89.42±0.13** | **92.87±0.24** | **92.36±0.18** |

[16], WRN [55], VGG [41], and MobileNet [39]), **(c)** settings (i.e., the teacher and the student share the architecture or use different architectures), **(d)** cases (i.e., regular cases, across-dataset cases, and data-limited cases). The competitors include FitNet [37], AT [56], SP [46], CC [35], PKT [31], AB [18], VID [2], RKD [30], CRD [45], SRRL [21], and CSKD [7]. Since CID uses the logits of teacher to provide context information for each sample, for a fair comparison, the KD [19] loss is added to all the competitors. We omit "+KD" for simplicity when denoting these competitors plus KD, e.g., "FitNet+KD" is abbreviated to "FitNet". On the other hand, since CID does intervention with the assistance of a linear layer, we also try to add an extra linear layer to the competitors, but we find that it hurts their performances due to overfitting, which we report in Appendix.

## 4.3 Model Compression

**CIFAR-10:** Table 1 reports the comparison results on CIFAR-10. We adopt the teacher and the student with the same architecture or different architectures. It is observed that CID consistently outperforms all the baselines significantly across different architectures on both settings, while there is no baseline consistently as the second best, since these baselines show their advantages in different architectures. These results demonstrate the superiority of CID.

**CIFAR-100:** We further report the comparison results on CIFAR-100 in Table 2. For a fair comparison, we adopt the architectures from the SOTA approaches (CRD [45] and SRRL [21]). As shown in Table 2, for compressing the large models to the smaller ones, CID obtains the best performances in different settings, which demonstrates the effectiveness of CID for model compression. The superior performances of CID are due to its ability to distill comprehensive knowledge and remove biases.

**Tiny ImageNet:** We further evaluate CID in more challenging datatset Tiny ImageNet. Table 3 shows that CID beats all the SOTA approaches substantially in terms of both Top-1 and Top-5 accuracies on the challenging dataset, which validates the usefulness and superiority of CID on different datasets.

**ImageNet:** To investigate the performance of CID on large scale datasets, we follow CRD by adopting ImageNet and using ResNet-34 and ResNet-18 as the teacher and the student, respectively. As shown

Table 2: Test accuracies (%) on CIFAR-100.

| Teacher (#Params) | WRN-40-4 (8.97M) | WRN-40-2 (2.26M) | ResNet-56 (0.86M) | ResNet-50 (23.71M) | ResNet-50 (23.71M) |
|---|---|---|---|---|---|
| Student (#Params) | WRN-16-2 (0.73M) | WRN-16-2 (0.73M) | ResNet-20 (0.29M) | MobileNetV2 (1M) | VGG-8 (4M) |
| Teacher | 79.50 | 75.61 | 72.34 | 79.34 | 79.34 |
| Vanilla Student | 73.26 | 73.26 | 69.06 | 64.60 | 70.36 |
| KD | 74.52±0.20 | 74.92±0.28 | 70.66±0.24 | 67.35±0.32 | 73.81±0.13 |
| FitNet | 74.48±0.27 | 75.12±0.33 | 70.70±0.24 | 66.96±0.24 | 73.24±0.27 |
| AT | 74.70±0.13 | 75.32±0.15 | 71.08±0.34 | 66.13±0.23 | 74.01±0.25 |
| SP | 74.79±0.31 | 74.98±0.28 | 70.66±0.12 | 68.54±0.35 | 73.52±0.25 |
| CC | 74.48±0.19 | 75.09±0.23 | 71.30±0.31 | 68.95±0.15 | 73.48±0.29 |
| VID | 74.83±0.10 | 75.14±0.15 | 71.18±0.09 | 68.34±0.31 | 73.46±0.25 |
| RKD | 74.66±0.26 | 74.89±0.20 | 70.93±0.25 | 68.66±0.34 | 73.51±0.33 |
| PKT | 75.21±0.22 | 75.33±0.18 | 71.53±0.26 | 68.41±0.14 | 73.61±0.28 |
| CRD | 75.49±0.28 | 75.64±0.21 | 71.63±0.15 | 69.54±0.39 | 74.58±0.27 |
| SRRL | 75.96±0.21 | 75.96±0.25 | 71.44±0.18 | 69.45±0.29 | 74.46±0.25 |
| CSKD | 74.66±0.35 | 75.11±0.15 | 71.30±0.26 | 68.80±0.36 | 73.61±0.17 |
| CID | **76.40±0.15** | **76.55±0.19** | **71.90±0.27** | **69.68±0.26** | **74.75±0.17** |

Table 3: Test accuracies (%) on Tiny ImageNet.

| | Teacher: WRN-40-2, Student: WRN-16-2 | | Teacher: VGG-13, Student: VGG-8 | |
|---|---|---|---|---|
| | Top-1 (%) | Top-5 (%) | Top-1 (%) | Top-5 (%) |
| Teacher | 61.84 | 84.11 | 61.62 | 81.71 |
| Vanilla Student | 56.13 | 79.96 | 55.46 | 78.15 |
| KD | 58.27±0.17 | 82.10±0.15 | 60.21±0.19 | 81.61±0.28 |
| FitNet | 59.58±0.24 | 82.59±0.18 | 60.11±0.13 | 82.11±0.16 |
| SP | 58.52±0.36 | 82.10±0.15 | 60.94±0.24 | 82.42±0.20 |
| CC | 60.12±0.12 | 83.08±0.10 | 61.11±0.34 | 82.44±0.28 |
| VID | 59.91±0.10 | 83.16±0.17 | 61.35±0.17 | 82.61±0.23 |
| RKD | 59.29±0.23 | 82.99±0.07 | 60.54±0.25 | 82.39±0.16 |
| CRD | 59.86±0.29 | 83.18±0.15 | 61.98±0.27 | 82.64±0.19 |
| SRRL | 59.90±0.25 | 82.98±0.21 | 61.30±0.21 | 82.31±0.26 |
| CID | **60.51±0.19** | **83.52±0.20** | **62.86±0.18** | **83.81±0.13** |

in Table 4, CID outperforms these competitors significantly, which demonstrates the applicability and effectivness of CID on large scale datasets.

## 4.4 Transferability Comparison

An important goal of representation learning is to learn general representations which can be transferred to different datasets. We investigate the across-dataset generalization ability of CID. For a fair comparison, we follow the settings of CRD. Specifically, we freeze the feature encoder of the student and train a linear classifier on STL-10 [9] or TinyImageNet. WRN-40-2 and WRN-16-2 are adpoted as the teacher and the student, respectively.

The transferability comparison results are reported in Table 5. It is clearly observed that CID beats the prior work substantially on both datasets, which demonstrates its superior generalization ability on new data. The reason is that when transferring the knowledge from one dataset to another, the inherited bias from the teacher can be a disaster to the new dataset. The ability of CID to remove the biased knowledge mitigates this issue, thus leading to a better generalization on new datasets.

## 4.5 Data-Limited Distillation Performances

In reality, it happens that when a powerful model is released, only a few data samples are publicly accessible due to the privacy or confidentiality issues in various domains such as medical and industrial

Table 4: Comparison results on ImageNet.

|  | Teacher | Vanilla Student | KD | OFD | AT | SRRL | CRD | SP | CC | CID |
|---|---|---|---|---|---|---|---|---|---|---|
| TOP-1 (%) | 73.3 | 69.8 | 70.7 | 71.1 | 70.7 | 71.7 | 71.4 | 70.2 | 70.0 | **71.9** |
| TOP-5 (%) | 91.4 | 89.1 | 89.9 | 90.1 | 90.0 | 90.6 | 90.5 | 89.8 | 89.2 | **90.7** |

Table 5: Transferability performances.

| Cross-dataset | Teacher | Student | KD | AT | FitNet | CRD | SRRL | CID |
|---|---|---|---|---|---|---|---|---|
| CIFAR-100 to STL-10 | 68.6 | 69.7 | 70.9 | 70.7 | 70.3 | 72.2 | 71.0 | **72.5** |
| CIFAR-100 to Tiny ImageNet | 31.5 | 33.7 | 33.9 | 34.2 | 33.5 | 35.5 | 34.3 | **35.9** |

Table 6: Comparison results in the data-limited scenario.

| Training Data | Student | KD | FitNet | SP | CC | RKD | PKT | CRD | SRRL | CID |
|---|---|---|---|---|---|---|---|---|---|---|
| 20% | 52.50 | 59.14 | 58.41 | 60.35 | 58.60 | 58.95 | 59.48 | 59.07 | 59.30 | **62.13** |
| 40% | 61.45 | 66.89 | 65.94 | 66.73 | 66.27 | 66.15 | 66.13 | 66.84 | 66.40 | **68.64** |
| 60% | 65.57 | 69.90 | 69.21 | 69.70 | 69.38 | 69.74 | 70.18 | 70.53 | 70.01 | **70.85** |

domains. It is thus necessary for distillation approaches to work on these practical cases. We compare CID with the existing approaches in the data-limited scenario on CIFAR-100 by using VGG-13 and VGG-8 as the teacher and the student, respectively. As shown in Table 6, CID outperforms all the baselines by a large margin in all the three cases with 20%, 40%, and 60% training data. We also notice that the advantage of CID is more obvious when fewer training data are available, e.g., the improvement of CID over the second best method is about 2% on 20% or 40% training data cases, which is much higher than the improvement on 60% training data. The reason is that when fewer data samples are available, these samples are severely inadequate to represent the real data distribution so that the biases become more serious. While the existing approaches fail to handle this issue, CID is able to address it with the interventional distillation, which leads to a better performance.

## 5  Conclusion, Limitations, and Broader Impact

**Conclusion.** In this paper, we have proposed comprehensive, interventional distillation (CID) that captures both sample and class representations while removing the bias by using softened logits as the context information based on causal intervention. To our best knowledge, CID is the first framework along the line of using causal inference to address KD-based model compression. To this end, CID is able to keep the good representations and remove the bad bias. Extensive experiments demonstrate that CID has a better generalization ability on test data and a better transferability across different datasets against the existing SOTA approaches.

**Limitations.** A major assumption in CID is that the training data used by the teacher and the student are from the same distribution. The assumption is typically satisfied in knowledge distillation literature as almost all the existing work uses the same data to train the teacher and the student. On the other hand, when the assumption is violated, new biases will be introduced from the new data. CID is not designed to solve this problem and we leave this question to the future work. Also, when the training data used by the student and the teacher differ substantially, the teacher may not be able to supervise the student anymore. The role of the teacher needs to be changed in this case, which we leave for the future work.

**Broader Impact.** There is an increasing interest in implementing DNNs on portable devices such as smart phones and watches, while DNNs need a large amount of memory and computation, which highly limits their deployments on these resource-limited devices. CID can be used to address this issue by compressing large models (teachers) to small and fast ones (students). The advantage of CID over the other distillation approaches is that it not only enables the students to inherit comprehensive knowledge from the teachers but also removes the bad biased knowledge, which leads to a better generalization and transferability. More essentially, in real world, collecting data is very expensive, while using sparse data points to train a student induces severe biases, which poses challenges to the existing distillation approaches. CID is able to address this problem with the interventional distillation. So far, no negative impact has been observed.

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
