# Appendix for "Comprehensive Knowledge Distillation with Causal Intervention"

## A  Implementation Details

**Datasets.**  CIFAR-10 is an image classification dataset. It contains 50,000 training images and 10,000 test images of 10 classes. We adopt the standard data augmentation strategy on CIFAR datasets, i.e., padding 4 pixels on each side of an image and randomly flipping it horizontally, and then cropping it to $32 \times 32$ size. CIFAR-100 comprises similar images to those in CIFAR-10, but has 100 classes. Tiny ImageNet is a subset of ImageNet. It has 200 classes with 100,000 training images and 10,000 test images. We adopt the standard data argumentation, i.e., padding 8 pixels on each side of an image and randomly flipping it horizontally, and then randomly cropping it to $64 \times 64$ size. ImageNet is a large-scale image classification dataset containing 1.28 million training images and 50,000 validation images of 1000 classes. The standard augmentation [6, 4] is adopted.

**Training Hyperparameters.**  The representation based approaches can benefit from combining with KD [2]. We add the KD loss to all the baselines for a fair comparison, since CID uses the softened logits to provide the context information for each image. The training objectives of these baselines can be written as:

$$\mathcal{L}_{baseline} = \mathcal{L}_{CE} + \mathcal{L}_{KD} + c\mathcal{L}_{distill} \tag{1}$$

where $\mathcal{L}_{CE}$, $\mathcal{L}_{KD}$, and $\mathcal{L}_{distill}$ are the regular cross-entropy loss, the KL-divergence loss in KD, and the specific distillation loss in the baseline, respectively. For all the baselines, we use the implementation of CRD [6] or the author-released implementation.

CID has two hyperparameters. To save the cost, we do a very basic search instead of grid search. We first fix $\beta$ and search for the best $\alpha \in [1, 70]$ with interval 5 to interval 1, and then we fix $\alpha$ and search for the best $\beta \in [0, 10]$. We follow the training setting in CRD [6] and SRRL [4]. The GPUs that we used include RTX-2080ti, TITAN V, Tesla-P100, and GTX-1080.

Table 1: Combining interventional distillation and baselines.

| Student | AT | AT+Intervention | SP | SP+Intervention | CC | CC+Intervention | RKD | RKD+Intervention |
|---|---|---|---|---|---|---|---|---|
| WRN-16-2 | 75.32 | **75.56** | 74.98 | **75.42** | 75.09 | **75.20** | 74.89 | 74.90 |
| ResNet-20 | 71.08 | **71.20** | 70.66 | **71.17** | 71.30 | 71.33 | 70.93 | **71.26** |

## B  Combining Interventional Distillation and Baselines

In this part, we investigate whether the proposed interventional distillation in CID benefits the existing approaches. For all the baselines, the original KD [2] loss is included. We omit "+KD" for simplicity when denoting "baseline+KD". We use two teacher-student pairs on CIFAR-100, i.e., WRN-40-2 and WRN-16-2, ResNet-56 and ResNet-20.

We report the performances in Table 1. It is observed that overall the baselines combining with the interventional distillation outperform the baselines significantly, which indicates that the proposed interventional distillation loss is compatible with these approaches.

35th Conference on Neural Information Processing Systems (NeurIPS 2021).

Table 2: Combining class representation distillation and baselines.

| AT | AT+ClassRep | SP | SP+ClassRep | CC | CC+ClassRep | RKD | RKD+ClassRep |
|---|---|---|---|---|---|---|---|
| 71.08 | **71.50** | 70.66 | **71.56** | 71.30 | **71.52** | 70.93 | **71.26** |

Table 3: More comparison results. For a fair comparison, all these approaches use the same pretrained teacher downloaded from CRD [6] except teacher WRN-40-4 that CRD does not provide. * denotes that we run the author-released code with the teacher downloaded from CRD.

| | | WRN-16-4 | ResNet-56 | ResNet-56 | WRN-16-4 | ResNet-110 |
|---|---|---|---|---|---|---|
| | Teacher | WRN-16-4 | ResNet-56 | ResNet-56 | WRN-16-4 | ResNet-110 |
| | Student | WRN-16-1 | ResNet-14 | ResNet-8 | ResNet-14 | WRN-16-1 |
| CIFAR-10 | Teacher | 95.04 | 93.87 | 93.87 | 95.04 | 94.00 |
| | Vanilla Student | 91.32 | 91.33 | 88.55 | 91.33 | 91.32 |
| | NST | 92.42 | 91.97 | 86.78 | 92.16 | 91.89 |
| | FT | 92.58 | 91.93 | 89.16 | 92.49 | 92.04 |
| | AB | 92.43 | 92.13 | 88.85 | 92.58 | 91.40 |
| | WKD* | 92.46 | 91.78 | 89.21 | 92.62 | 91.73 |
| | CID | **92.95** | **92.31** | **89.42** | **92.87** | **92.36** |
| | Teacher | WRN-40-4 | WRN-40-2 | ResNet-56 | ResNet-50 | ResNet-50 |
| | Student | WRN-16-2 | WRN-16-2 | ResNet-20 | MobileNetV2 | VGG-8 |
| CIFAR-100 | Teacher | 79.50 | 75.61 | 72.34 | 79.34 | 79.34 |
| | Vanilla Student | 73.26 | 73.26 | 69.06 | 64.60 | 70.36 |
| | NST | 75.05 | 74.67 | 71.07 | 68.06 | 71.74 |
| | FT | 74.56 | 75.15 | 70.35 | 64.13 | 72.98 |
| | AB | 74.68 | 70.27 | 71.68 | 67.88 | 73.65 |
| | WKD* | 74.80 | 75.29 | 71.42 | 67.20 | 73.39 |
| | CID | **76.40** | **76.55** | **71.90** | **69.68** | **74.75** |

## C Combining Class Representation Distillation and Baselines

We further study whether the proposed class representation distillation benefits the existing approaches. We adopt ResNet-56 and ResNet-20 as the teacher and the student on CIFAR-100, respectively.

The results are reported in Table 2. The class representation distillation improves the performances of these baselines substantially, which demonstrates its effectiveness and generalization across different approaches.

## D More Comparison Results

In this section, we report comparison results with more baselines including NST [3], FT [5], AB [1], and WKD [7]. Note that WKD used a different (re-pretrained) teacher in the original paper from these baselines. For a fair comparison, we run the author-released code and use the same teacher downloaded from CRD [6]. The comparison results are reported in Table 3. CID consistently beats these baselines significantly across different architectures and datasets, which validates its effectiveness and superiority.

## E Effects of Hyperparameters

We report how the performance of CID varies with the two hyperparamters $\alpha$ and $\beta$ in CID in Table 4, where we adopt WRN-40-2 and WRN-16-2 on CIFAR-100 as the teacher and the student, respectively. As expected, the sample representation weight $\alpha$ has more influence on the performance than $\beta$, since sample representations directly involve in making the final prediction, and are further used in class representation distillation and interventional distillation.

Table 4: Effects of hyperparameters.

|  | $\alpha$=5 | $\alpha$=10 | $\alpha$=20 | $\alpha$=30 | $\alpha$=50 |
|---|---|---|---|---|---|
| $\beta$=0.05 | 75.20 | 75.38 | 75.72 | 75.49 | 76.10 |
| $\beta$=0.1 | 75.58 | 75.94 | 76.00 | 76.01 | 76.00 |
| $\beta$=1 | 75.78 | 75.95 | 76.01 | 76.03 | 76.55 |
| $\beta$=10 | 75.41 | 75.76 | 76.07 | 76.34 | 75.99 |

Table 5: Baselines with an extra linear layer on CIFAR-100.

|  | FitNet | FitNet+Linear | AT | AT+Linear | SP | SP+Linear | CC | CC+Linear |
|---|---|---|---|---|---|---|---|---|
| Acc | **74.48** | 74.08 | **74.70** | 74.44 | **74.79** | 73.72 | **74.48** | 73.79 |

|  | VID | VID+Linear | PKT | PKT+Linear | NST | NST+Linear | RKD | RKD+Linear |
|---|---|---|---|---|---|---|---|---|
| Acc | **74.83** | 74.24 | **75.21** | 74.81 | **75.05** | 74.96 | **74.66** | 74.04 |

# F   Baselines with an Extra Linear Layer

CID does interventional distillation with the assistance of a linear layer. For a fair comparison with the baselines, we try to add an extra linear layer to the baselines. We find that the performances of the baselines decrease with an extra linear layer due to overfitting.

We report the results in Table 5, where we adopt WRN-40-4 and WRN-16-2 as the teacher and the student, respectively. We observe that an extra linear layer hurts the performances of the baselines largely, which may be due to overfitting. This indicates that the performance gain of CID is not from the few parameters in the linear layer but from the causal intervention distillation which pursues the true causality from $X$ to $Y$.