# OpenReview forum: "Comprehensive Knowledge Distillation with Causal Intervention"
_NeurIPS.cc/2021/Conference — NeurIPS 2021 Poster_

### Official Review · Reviewer_3nmW · 2021-07-13

**Rating:** 6
**Confidence:** 3

**Summary:**

This paper proposes an improved knowledge distillation process named Comprehensive, Interventional Distillation (CID). It adds a class representation term to the loss function of a knowledge distillation model to enforce the student and the teacher to learn not only similar sample representations but also class representations, where a class is represented by the similarity matrix among all of its samples. It further modifies the loss function term of the student to remove the bias learned by the teacher from the context information in training data, based on casual intervention. Experimental results on four real datasets confirm the effectiveness of the proposed model in improving the classification accuracy of the student model.

**Limitations And Societal Impact:**

The experimental results are not very strong. The claim "the performances drop significantly without anyone of these terms" for Fig. 6 seems to be an overstatement. The performance drop for w/o CR and w/o INV are just about 0.5%. Similar claims appear in the discussions on the rest of the experimental results such as "outperforms all the baselines significantly", "beats all the SOTA approaches substantially", "outperforms these competitors significantly", while the performance difference between the proposed model CID and the best baseline is less than 1% in many cases. What is the definition of "outperforms significantly/substantially"?

Given the claim "Since the bias knowledge in the teacher is caused by the training data, we assume that the training data used by the teacher and those used by the student are from the same distribution", is the bias induced by the context learned by the teacher still a problem?

Are there concrete examples in the test datasets that cause classification errors because of the context bias learned by the teacher model in the baseline models, which are correctly classified by the proposed CID model?

What is the implication in terms the extra training introduced by the CID model?

Typo: "but also server as a tool"

**Main Review:**

This paper studies an interesting problem, i.e.,  to transfer class representations and to remove biases with causal intervention in knowledge distillation. The paper is motivated with a figure (Figure 1) to highlight the importance of removing biases. However, it then makes an assumption that " the training data used by the teacher and those used by the student are from the same distribution". The importance of removing biases under such an assumption needs further justification.

Removing biases with causal intervention in knowledge distillation is new and interesting. The derivation of Equations (4) - (7) is  solid. The idea of learning and transferring class representations is also interesting but this is mainly a heuristic without performance guarantees.

The experiments are run on multiple real datasets of different sizes which is also good.

The paper is well written overall and is an interesting exploration. There are a few issues that need to addressed to further strengthen the paper as detailed in the "Limitations And Societal Impact" section.

Post-rebuttal:

I appreciate the authors' efforts to address my comments. I am not fully convinced by the response to Q2, as I am still unclear why learning the bias by the teacher is an issue given the assumption that the training data used by the teacher and those used by the student are from the same distribution (i.e., they may share the same bias). Since this is a basic assumption of the paper, I cannot boost my score.

And update after further author response:

I appreciate the further discussion, which clears the confusion. No further question or objection to acceptance from my end.


**Time Spent Reviewing:**

4 hours

---

> ### Author Response · Authors · 2021-08-09
> **Response to Reviewer 3nmW**
>
> Thank you for the valuable comments and the interests in the addressed problem and the idea. We clarify the concerns below.
>
> Q1: **“The experimental results are not very strong ... What is the definition of "outperforms significantly/substantially"?”** \
> A1: CID statistically significantly outperforms all the competitors consistently on different pairs of students and teachers across different datasets in different settings while **there is not a method consistently second-best**. Thus, we state “significantly/substantially" compared to the other methods (In Figure 6, CID also shows statistically significant improvements). Sorry for not defining the adverbs clearly and we will add a note to make it clear or we will revise the statement if the case is an overstatement.
>
> Q2: **“Given the claim "Since the bias knowledge in the teacher is caused by the training data, we assume that the training data used by the teacher and those used by the student are from the same distribution", is the bias induced by the context learned by the teacher still a problem?”** \
> A2: When the training data contain bias context, a teacher which is trained on these data with a regular cross-entropy loss also learns the bias. On the other hand, when the teacher is trained with a fancy approach that can remove the bias, the teacher may not contain the bias. The proposed method is to address the bias issue for the teacher in the first case which is also the common case in reality due to the high cost and difficulties for collecting data in different regions or settings.
>
> Q3: **“Are there concrete examples in the test datasets that cause classification errors because of the context bias learned by the teacher model in the baseline models, which are correctly classified by the proposed CID model?”** \
> A3: The answer is Yes. Empirically, we observe that some samples are corrected by CID and the ablation study also shows quantitative improvements.
>
> Q4: **“What is the implication in terms the extra training introduced by the CID model?”** \
> A4: As stated in Line 242-244 on Page 7 in the paper, CID does intervention with the assistance of a linear layer to learn the context information from the teacher. According to our experiments, a simple linear layer almost does not introduce training time or memory overhead.
>
> Q5: **"Typo: "but also server as a tool"** \
> A5: Thank you for pointing out this typo. We will correct it.

---

> ### Author Response · Authors · 2021-08-18
> **Response to Post-Rebuttal of Reviewer 3nmW**
>
> Thank you for reading the rebuttal and giving valuable feedback. We would like to further make this point clear.
>
> Q1: **"I am still unclear why learning the bias by the teacher is an issue given the assumption that the training data used by the teacher and those used by the student are from the same distribution (i.e., they may share the same bias)"** \
> A1: The proposed question can be converted to "if the teacher and the student learn the same bias from the same training data, is the bias still an issue?" The answer is Yes. The reason is that the context prior bias is **only learned from the training data** while the training data are typically **not comprehensive enough** to well represent the real context distribution due to the huge cost of collecting samples and labels in different settings in reality, i.e., **the test data typically have a different context prior from the training data in reality.** The learned prior bias thus can mislead the learned model to make wrong predictions based on the learned prior bias. For example, when almost all **the dogs in the training data are in grasses and the cats are within rooms,** the learned prior bias could mislead the classifier to classify **the dogs within rooms** in the test data as cats, and **the cats in grasses** in the test data as dogs. Therefore, even if the student and the teacher have learned the same bias, the bias can hurt the generalization on test data.
> \
> Second, the reason why we need the assumption that the training data used by the teacher and those used by the student are from the same distribution is that we need to know what kind of data (i.e., priors) cause the bias in the teacher knowledge and then we can remove the prior bias knowledge with causal intervention when distilling the knowledge into the student.

---

### Official Review · Reviewer_YwMj · 2021-07-14

**Rating:** 6
**Confidence:** 4

**Summary:**

Authors have proposed a novel framework (CID) for knowledge distillation that is able to capture sample and class representations and reduce knowledge bias induced by the context prior through causal intervention. There are two major differences between the proposed approach and previous work: first, CID is able to transfer class representations. Second, it uses soften logits as sample context information. Authors have provided extensive experimental results to show the effectiveness of their approach.

**Limitations And Societal Impact:**

Authors have adequately discussed limitations and societal impacts.

**Main Review:**

The paper is well-written and easy to follow. The idea is novel although there has been some usage of causal inference for knowledge distillation for the question answering application. The major concern that I have is about the experimental results. Some of the cited papers that are recently published in top venues (e.g.  the SSKD approach: Knowledge distillation meets self-supervision from ECCV 2020) are not included in the tables. In addition, there are discrepancies  between numbers reported here vs. numbers reported in previous works. As an example, the performance of AT is reported 71.78 in the Table 3 of SSKD paper for (ResNet-56, ResNet-20); while it is reported as 71.08±0.34 in the submitted work. Also, there are some (teacher, student) configurations that are missing in this paper compared to the previous works.

I am more than happy to increase my score if authors can provide a fair explanation about issues in the experimental results.

----------------
Post-rebuttal:

Thanks to authors for addressing my concerns and questions regarding experimental results. I have updated my score.

**Time Spent Reviewing:**

5

---

> ### Author Response · Authors · 2021-08-09
> **Response to Reviewer YwMj**
>
> Thank you for the valuable comments and recognizing the idea. We address the experimental concerns below.
>
> Q1: **“Some of the cited papers that are recently published in top venues (e.g. the SSKD approach: Knowledge distillation meets self-supervision from ECCV 2020) are not included in the tables. ”** \
> A1: SSKD is not included in Table 2, since it is an unfair comparison for the other methods. SSKD uses strong data augmentation (e.g., rotation) in the self-supervised tasks, which means that SSKD sees more data than the other methods. SSKD also sees more tasks (self-learning tasks) than the other methods, which means that it is along a different line from the other methods that uses the standard supervised learning. Data augmentation and task augmentation are also complementary to the other methods. Nevertheless, CID still outperforms SSKD as shown below, where the results of SSKD are cited from the SSKD paper: \
> SSKD: WRN-40-2 to WRN-16-2: 76.04; Resnet56 to ResNet20: 71.49 \
> CID:    WRN-40-2 to WRN-16-2: 76.55; Resnet56 to ResNet20: 71.90.
>
> Q2: **“In addition, there are discrepancies between numbers reported here vs. numbers reported in previous works. As an example, the performance of AT is reported 71.78 in the Table 3 of SSKD paper for (ResNet-56, ResNet-20); while it is reported as 71.08±0.34 in the submitted work.”** \
> A2: Our results of the baselines are cited from CRD [\*] as we use their released (public) pretraind teachers. The baseline differences from the SSKD [#] paper result from the pretrained teachers as SSKD uses their own pretrained higher-accuracy teachers. For reproducibility and a fair comparison, we use the widely-used, publicly available pretrained teachers for all the baselines which are released from CRD. Since SSKD uses higher-accuracy pretrained teachers than CRD (e.g., the accuracy of the pretrained teacher WRN-40-2 in CRD is 75.61 while that in SSKD is 76.46), the students in the baselines also have higher accuracy than those in CRD. \
> [\*] Tian, Yonglong, et al.. "Contrastive representation distillation." In International Conference on Learning Representations, 2020. (Our baselines are mainly cited from Table 7 on Page17.) \
> [#] Xu, Guodong, et al. "Knowledge distillation meets self-supervision." European Conference on Computer Vision. Springer, Cham, 2020.
>
> Q3: **“Also, there are some (teacher, student) configurations that are missing in this paper compared to the previous works.”** \
> A3: As we need to report the experiment results in different settings (e.g., transferability and data-limited cases) but there is a space limitation, we only report five pairs of students and teachers on CIFAR-10 and CIFAR-100. These results on five different pairs have already demonstrated the effectiveness of CID. Now we add more pairs of students and teachers (on CIFAR-100) below.
>
> |                                                                | KD         |  FitNet  | AT     | CC     | CRD    | CSKD | CID |
> | :---:                                                          | :-:          | :-:          | :-:      | :-:      | :-:        | :-:         | :-:    |
> | resnet-110 to resnet-20                            | 70.67     | 70.67    | 70.97 | 70.88 | 71.56 | 70.82 | **71.88** |
> | resnet32$\times$4 to resnet8$\times$4 | 73.33     | 74.66    | 74.53 | 74.21 | 75.46 | 74.37 | **75.85** |

---

### Official Review · Reviewer_BQzY · 2021-07-16

**Rating:** 6
**Confidence:** 5

**Summary:**

This paper focuses on comprehensive knowledge distillation to transfer the class representation. The proposed method aims to improve the performance of the student model by considering the transferring of class representation. It is a novel view, especially with a causal intervention approach. The experiments are conducted to demonstrate the advantages of the proposed method.

**Limitations And Societal Impact:**

- the motivation of the interventional distillation is unclear. What is the confounder or bias that affected the comprehensive distillation? How to guarantee the prior knowledge K is not the same as the How to ensure that the prior knowledge K in interventional distillation and knowledge in comprehensive representation distillation are not in conflict.
- how to know the prior knowledge K is a confounder of X and Y? it is against our intuition. Which type of prior knowledge?
- in the introduction, the authors claim that "good class representations are beneficial to sample representation learning, since they can shape the sample representation distribution." It is hard to understand. How do the class representations shape the sample representation distribution?
-the idea of considering class representation or category structure into knowledge distillation is not novel. Please compare with the work with Chen. et.al.
Chen et.al., Improving Knowledge Distillation via Category Structure. ECCV 2020.

**Main Review:**

Considering class representation or category structure into knowledge distillation is not a novel approach to improve the performance of knowledge  distillation. There are some works done with the same idea such as Chen et.al.  But the introducing of causal intervention is novel in knowledge distillation.  The writing and organization of this paper are clear, and the motivation of comprehensive representation distillation is also meaningful and significant.  The experiments and analysis are sufficient.

Post-rebuttal:

Many thanks to the authors for addressing my concerns. After reading the rebuttal, I have updated my score.

**Time Spent Reviewing:**

8

---

> ### Author Response · Authors · 2021-08-09
> **Response to Reviewer BQzY**
>
> Thank you for the valuable comments. We address the questions below.
>
> Q1: **“the motivation of the interventional distillation is unclear”** \
> A1: The motivation for the interventional distillation is directly given in Paragraph 3 in Introduction and Figure 1 (as well as Section 3.2.1 and Figure 4). We restate the motivation as follows: the existing approaches enforce the student to fully imitate the teacher while ignoring the fact that the teacher is typically not perfect and contains unignorable bias knowledge which is usually induced by the context prior (e.g., background) in the training data as illustrated in Figure 1. We then propose to use causal intervention to address this issue.
>
> Q2: **“What is the confounder or bias that affected the comprehensive distillation?”** \
> A2: (a) The confounder or bias prior is directly given in Section 3.2.1 and Figure 4 (as well as Figure 1 and Paragraph 3 in Introduction). For instance, when the dogs in the training dataset are usually on green grasses and the cats are within rooms, it can mislead the trained classifier to classify the cats on green grasses in the test dataset as dogs, and the dogs within rooms as cats due to the bias induced by the context. \
> (b) There is a misunderstanding that “bias that affected the comprehensive distillation”. In fact, the bias does not affect the comprehensive distillation. The comprehensive distillation without causal intervention transfers both the good knowledge and the bias (prior) knowledge to the student. We aim to remove the inherited bias knowledge from the comprehensive knowledge with causal intervention.
>
> Q3: **“How to guarantee the prior knowledge K is not the same as the How to ensure that the prior knowledge K in interventional distillation and knowledge in comprehensive representation distillation are not in conflict.”** \
> A3: There is no conflict between comprehensive knowledge and the interventional distillation, since the interventional distillation is to remove the bias prior knowledge from the comprehensive knowledge. Therefore, the bias knowledge is actually a part of the comprehensive knowledge and the interventional distillation aims to remove its bad effects.
>
> Q4: **“how to know the prior knowledge K is a confounder of X and Y? it is against our intuition. Which type of prior knowledge?”** \
> A4: The reason why K is the confounder of X and Y is directly given in Section 3.2.1 and Figure 4. Specifically, besides the regular and intuitive X → Y , the prediction is also affected by the prior knowledge K. For example, in Figure 1, the cats in the test dataset are misclassified to dogs, since the context prior in K misleads the model to focus on the grass feature in X. We can see that the prior K indeed affects the prediction Y. On the other hand, the context prior in K also determines where the object appears in an image X, e.g., the context prior in the training dataset in Figure 1 puts the dog object in green grasses instead of rooms. Therefore, K is the confounder of X and Y.
>
> Q5: **“in the introduction, the authors claim that "good class representations are beneficial to sample representation learning, since they can shape the sample representation distribution." It is hard to understand. How do the class representations shape the sample representation distribution?”** \
> A5: We define the class representation as a graph over a class. Thus, the class representation shapes how the representations of the samples in a class distribute. Consequently, a good class representation can benefit sample representation distribution. We have also empirically demonstrated the benefits of class representations in the ablation studies, i.e., Figure 6 in the paper and Table 2 in Appendix.
>
> Q6: **“the idea of considering class representation or category structure into knowledge distillation is not novel. Please compare with the work with Chen. et.al. Chen et.al., Improving Knowledge Distillation via Category Structure. ECCV 2020.”** \
> A6: The comparison between the mentioned work (CSKD) and our method CID has already been given in Table 2 in the paper (Due to the space limitation, we previously only reported the results on this dataset). It is observed that CID performs much better than CSKD. The advantages (or differences) of CID over CSKD is that (a) the sample-class relation defined in CSKD only captures sample-to-class-center similarity while failing to capture sample-to-sample relations and thus cannot well represent a class; (b) CSKD fails to consider the bias knowledge in the representations from the teacher while CID removes the bias knowledge with causal intervention. In addition to the results in Table 2, we add more comparison results below, which is conducted on Tiny ImageNet:
>
> |           | WRN-40-2   to     | WRN-16-2     |       | VGG-13 to    | VGG-8|
> | :---:     | :-:             | :-:                              | :-:   |  :-:                 | :-:       |
> |           | Top-1        | Top-5                         |        |  Top-1        | Top-5
> | CSKD | 59.87       |   82.90                        |        |  61.32        | 83.09 |
> | CID     | **60.51** |  **83.52**                   |         |  **62.86**    |  **83.81** |

---

### Official Review · Reviewer_tHsZ · 2021-07-16

**Rating:** 6
**Confidence:** 3

**Summary:**

This paper proposes a new knowledge distillation method called CIS that captures both sample and class representations from the teacher while removing the bias with causal intervention. Experiments on several benchmark datasets validate the effectiveness of the proposed method.


**Ethical Concerns:**

No.

**Limitations And Societal Impact:**

This paper has addressed the limitations and potential negative societal impact of their work.

**Main Review:**

Strengths:
1. This paper is well written and easy to follow.
2. The motivation is clear. I think it is the first attempt to investigate the bias problem in teacher's prediction.
3. Experiments are done on different benchmarks, and comparison covers SOTA methods.

Weakness:
The class representation part is unclear to me. As is described in (2). The class representation dimension is related to the number of images in the current category. So it will inevitably increase the training time. However, this paper did not provide training time in the current version. Besides, the class representation idea is similar to [33]. Although [33] only considers within batch comparison, but it considers the inter-class comparison while this paper only considers the intra-class comparison. The advantage over [33] should be discussed and analysed.

Post-rebuttal:

Thanks to the authors for addressing my concerns and questions regarding the class representation part. I keep my previous score.


**Time Spent Reviewing:**

1hour

---

> ### Author Response · Authors · 2021-08-09
> **Response to Reviewer tHsZ**
>
> Thank you for the valuable comments and recognizing the motivation. We address the concerns below.
>
> Q: **“The class representation part is unclear to me. As is described in (2). The class representation dimension is related to the number of images in the current category. So it will inevitably increase the training time. However, this paper did not provide training time in the current version. Besides, the class representation idea is similar to [33]. Although [33] only considers within batch comparison, but it considers the inter-class comparison while this paper only considers the intra-class comparison. The advantage over [33] should be discussed and analysed.”**
>
> A: Sorry for the confusion. The class representation is defined as a large graph over the whole class, while at the implementation level, we cannot directly optimize this graph due to its large size. Instead, in each batch, we optimize a sub-graph of this large graph by using the samples in this class in the current batch. The difference between the class representation in CID and [33] is discussed in the related work that [33] ignores the class information by directly transferring the sample correlation over the whole dataset to the student, which contains redundant and irrelevant information as pointed out in [7]. [33] thus cannot learn a good class representation as it tries to learn all the complex and redundant relations over the whole dataset and loses focus. The proposed class representation is for shaping the sample representation distribution in a class but not for capturing the inter-class differences. The class differences are captured by the classification loss (i.e., $\mathcal{L}_{inv}$) in CID as the representations are fed into the classifier for discriminating the samples from different classes. The experimental results also validate the superior performances of CID over [33].

---

### Official Review · Reviewer_6Byr · 2021-07-18

**Rating:** 6
**Confidence:** 4

**Summary:**

The paper addresses an interesting concern : how can we transfer knowledge from a large model (master) to a tiny one (student) taking into account possible bias in the learning process of the teacher
The framework considered is that of  image classification, where the context contained is the image can yield some bias in the classification task (for e.g. garden when distinguishing a dog from a cat)
The paper considered two kinds of knowledge transfer
* one may want to transfer representation of each class
* one may also want to transfer the classifier in itself

The approach proposed in this paper to circumvent the bias problem is based on causal  intervention.  The claim is that there is an underlying graph where the output of the classifier may have as parent in the graph, both the true representation of the object to identify and the context
The idea is to perform causal intervention to remove the influence of the context in the classification
Numerical experiments are performed in the image context. The task is object clasiffication and several classical datasets are considered

**Ethical Concerns:**

OK from this point of view

**Limitations And Societal Impact:**

OK from this point of view

**Main Review:**

* I am quite convinced by the relevance of the problem which is addressed
* The approach in itself : first separation of the two kinds of knowledge that we want to transfer and second use of do calculus to remove influence of the context in classification in the teacher model is smart. I like it
* I wonder how difficult the objective function defined through do calculus is difficult to solve. I have no idea of possible properties of this function. It is probably non convex, I don't know if it is smooth. Can you give some details about that?
* In numerical experiments, the compression rate is significant and the results seems better than other state-of-art approaches. Results are mainly given hen the teacher is a ResNet or a WRN. Other architectures are mentionned but the resulats are not given. Can you give some details?
* What about robustness to noise of the approach? Are the performance of the student similar if the images are noisy or if there is noisy labels? One can think context may help when the images are mislabeled. Is there any drawback of this approach? What do we miss when we remove the impact of context?


**Time Spent Reviewing:**

3h

---

> ### Author Response · Authors · 2021-08-09
> **Response to Reviewer 6Byr**
>
> Thank you for the valuable comments and liking the motivation and idea. We address the concerns below.
>
> Q1: **"I wonder how difficult the objective function defined through do calculus is difficult to solve ... Can you give some details about that?"** \
> A1: The properties of the do-calculus-loss in CID are similar to those of the regular cross-entropy loss. The difference lies in the features that are fed into the loss. CID considers the context to remove the bias when extracting the features, while it does not change the function properties
>
> Q2: **"Results are mainly given when the teacher is a ResNet or a WRN. Other architectures are mentioned but the results are not given. Can you give some details?"** \
> A2: We use ResNet and WRN (plus VGG) as the teachers for two reasons: (1) teacher models need to be large and powerful enough to teach a student. ResNet and WRN (plus VGG) can be stacked to be very deep and large to be the teacher, e.g., 40 layers or 50 layers, while the other architectures such as MobileNetV2 are light-weight models so that they serve as the students instead of teachers; (2) for a fair comparison, the adopted teacher-student pairs are cited from the existing approaches CRD and SSRL. The results involving the other architectures such as MobileNetV2 and VGG are reported in Table 2 and Table 3 in the paper.
>
> Q3: **“What about robustness to noise of the approach? Are the performance of the student similar if the images are noisy or if there is noisy labels? One can think context may help when the images are mislabeled. Is there any drawback of this approach? What do we miss when we remove the impact of context? ”** \
> A3: (a) When the labels are noisy, the powerful teacher can simply correct them just like how the mean-teacher works in semi-supervised learning. When the input images are noisy, the proposed CID has advantages over the existing approaches, since it pursues the true causality from the object to the label by removing the object-unrelated bias. We add an experiment to verify this point by injecting Gaussian noise to 50% of the training images and keeping the other 50% of the training images clean. Our method CID performs better than the baselines as shown below.
>
> |                                           | FitNet  |  AT     | SP    | CC     | CRD   | CSKD |  CID      |
> | :---:                                    |     :-:      |    :-:   |:-:       | :-:       |:-:        | :-:        | :-:          |
> | resnet56 to resnet20         | 68.65   | 68.63 | 68.45 | 68.82 | 69.21 | 69.08 | **69.50** |
> | Resnet50 to MobileNetV2 | 64.52   | 64.05 | 65.58 | 65.93 | 66.38 | 65.87 | **66.64** |
>
> (b) The context can help the mislabeled images only if the unseen (test) images have the same context distribution as that of the training data. However, in reality, due to the huge cost of collecting samples and labels in different settings, the training data are not “comprehensive enough” to well represent the context distribution. Thus, the context in the training data typically introduces harmful bias priors and hurts the generalization on test data. \
> (c) The drawback of CID or what we miss after removing the context prior is that when the training data have the same context distribution as that of the test data, it fails to use the context to make predictions. For example, when the dogs in the training data and test data are all in the grass and the cats are all within rooms, one can use the context, e.g., grasses or rooms, to make predictions. However, in reality, limited to the data collection processes or regions, training data and test data typically have context discrepancies.

---

### Decision · Program_Chairs · 2021-09-27

**Decision:**

Accept (Poster)

**Comment:**

The paper is concerned with the distillation / compression of a large teacher model. Besides the transfer of the sample representation, the originality of the approach lies in the transfer of the class representation, and the use of a causal intervention (CID) aimed to prevent the transfer of the teacher's biases to the student (that can indeed be harmful even when the training and test distributions are the same).

Formally, CID proceeds by aligning $P(outcome | do (input))$ for the teacher and student, using the backdoor adjustment formula to account for the confounders (context of the different classes).

The merits of the approach are supported by extensive empirical comparisons with the state of the art, considering various architectures and datasets, and the relative importance of the three aspects (transferring the sample representation, transferring the class representation and using interventions to prevent the transfer of the biases) is shown using lesion studies. It is a bit disappointing that the interventional term, that is the most innovative and theoretically grounded one, is found to be the less important one after the lesion studies.

The authors did a very good job of answering the reviews, justifying the choice of the considered architectures, examining the impact of noise, clarifying the difference w.r.t. related works (e.g., Improving Knowledge Distillation via Category Structure) and  explaining the discrepancies with the results reported for the baselines in previous papers, depending on the use of pre-trained models.

In summary, this paper tackles a relevant issue in an innovative way, and it is viewed as above the acceptance threshold by all reviewers.